# The MAL Family of Proteins: Normal Function, Expression in Cancer, and Potential Use as Cancer Biomarkers

**DOI:** 10.3390/cancers15102801

**Published:** 2023-05-17

**Authors:** Leticia Labat-de-Hoz, Armando Rubio-Ramos, Isabel Correas, Miguel A. Alonso

**Affiliations:** 1Centro de Biología Molecular Severo Ochoa, Consejo Superior de Investigaciones Científicas, Universidad Autónoma de Madrid, 28049 Madrid, Spain; llabat@cbm.csic.es (L.L.-d.-H.); arubio@cbm.csic.es (A.R.-R.); icorreas@cbm.csic.es (I.C.); 2Department of Molecular Biology, Universidad Autónoma de Madrid, 28049 Madrid, Spain

**Keywords:** prognosis, MARVEL domain superfamily, DNA methylation, copy number alterations, breast cancer, kidney cancer, lung cancer, pancreatic cancer, thymoma, endometrial cancer

## Abstract

**Simple Summary:**

The use of biomarkers can provide information about the outcome of cancer patients, the type of tumor, its malignant potential and the risk of recurrence, and may also be useful for guiding the selection of the optimal clinical treatment for the patient. There is sparse published information about the expression of *MAL* gene family members in cancer and their use as cancer biomarkers. On the other hand, large-scale studies have generated datasets that are freely available from public resources and that allow the expression of almost every human gene in multiple types of cancer to be analyzed. This review uses these two sources to investigate the expression of the *MAL* gene family in cancer and to explore their potential biomedical applications as cancer biomarkers.

**Abstract:**

The MAL family of integral membrane proteins consists of MAL, MAL2, MALL, PLLP, CMTM8, MYADM, and MYADML2. The best characterized members are elements of the machinery that controls specialized pathways of membrane traffic and cell signaling. This review aims to help answer the following questions about the *MAL*-family genes: (i) is their expression regulated in cancer and, if so, how? (ii) What role do they play in cancer? (iii) Might they have biomedical applications? Analysis of large-scale gene expression datasets indicated altered levels of *MAL*-family transcripts in specific cancer types. A comprehensive literature search provides evidence of *MAL*-family gene dysregulation and protein function repurposing in cancer. For *MAL*, and probably for other genes of the family, dysregulation is primarily a consequence of gene methylation, although copy number alterations also contribute to varying degrees. The scrutiny of the two sources of information, datasets and published studies, reveals potential prognostic applications of *MAL*-family members as cancer biomarkers—for instance, *MAL2* in breast cancer, *MAL2* and *MALL* in pancreatic cancer, and *MAL* and *MYADM* in lung cancer—and other biomedical uses. The availability of validated antibodies to some MAL-family proteins sanctions their use as cancer biomarkers in routine clinical practice.

## 1. Introduction

The rise of new technologies has allowed the development of large-scale cancer genomic initiatives, such as The Cancer Genome Atlas (TCGA, cancer.gov/tcga, accessed on 24 February 2023). These have generated large genomic and molecular profiling datasets of a wide range of cancers that may be freely accessed by the scientific community. Simultaneously, a growing number of web-based, user-friendly resources available for downloading, visualizing, and exploring these datasets have been developed. The ultimate goal of this tremendous effort is to improve our understanding of cancer by investigating the expression of every human protein-coding gene in multiple types of cancer. Detailed analysis of the transcriptomic datasets reveals, for instance, the level of expression of a given gene and whether its expression can predict patient survival in the most frequent types of cancer. In the case of existence of validated antibodies, this information is being completed for a small number of proteins by the immunohistochemistry (IHC) of arrays of histological sections from normal and cancer tissues. In this review, using datasets from public genomic resources, we have analyzed the expression of the seven genes that constitute the *MAL* gene family in distinct type of cancers and investigated their potential use as prognostic biomarkers. We have also revised the existing literature concerning *MAL*-family genes in cancer, with particular emphasis on the regulation of their expression, copy number variation, mutation, function, protein expression studies, and potential use as prognostic biomarkers.

## 2. The MAL Protein Family

### 2.1. Members and Structure

The MAL family of integral membrane proteins was first defined, although not comprehensively, more than 25 years ago [1,2]. The completion of the sequencing of the human genome allowed all other members of the family to be identified and a membrane-tetraspanning domain to be defined. The latter was called the MARVEL (MAL and related proteins for vesicle formation and membrane link) domain, present in all the members of the family and in other protein families [3]. The MARVEL superfamily includes, in addition to the MAL family, the CMTM (chemokine-like factor MARVEL transmembrane domain-containing), the Physin, and the TAMP (tight junction-associated MARVEL proteins) families (Figure 1A).

MAL is the founding member of the MAL family of proteins (Figure 1B). In addition to MAL, the MAL family includes six more members that are organized into three branches: a first branch with MAL, MAL2, and MALL (MAL-like); a second branch with plasmolipin (PLLP) and CMTM8, the latter of which connects the MAL family with the CMTM family; and a third branch with myeloid differentiation-associated marker (MYADM) and MYADM-like2 (MYADML2), which are the largest members of the MAL family. All these proteins contain one MARVEL domain, except MYADM and MYADML2, which contain two. Consistent with these two types of proteins, the Alphafold algorithm (alphafold.ebi.ac.uk, accessed on 24 February 2023) predicts four membrane-spanning segments for MAL, MAL2, MALL, PLLP, and CMTM8, and eight similar segments for MYADM and MYADML2 (Figure 1C).

In addition to the seven genes that make up the *MAL* gene family, a locus called *MYADML* or *MYADML1*, which predicts a protein similar to MYADM and MYADML2 but lacks the first two transmembrane domains, is considered a pseudogene and will not be mentioned further.

### 2.2. Biochemical Features

The first differential feature reported for some MAL-family proteins is that they are highly soluble in the organic solvents commonly used to extract cell lipids, unlike most of the membrane proteins, which are excluded [4,5,6]. This biochemical behavior allows them to be included in the proteolipid group [7], together with a few proteins displaying similar lipid-like properties, such as the membrane-tetraspanning myelin proteolipid protein (PLP) and the 16-kDa subunit of the V_o_ sector of the eukaryotic H^+^-pump V-ATPase, which are among the best known representatives [8]. An intriguing property of PLP, shared by PLLP and probably by other related proteolipids, is that it can be converted in vitro into a water-soluble form by gradually replacing the organic solvent by water [9,10,11]. This unusual behavior raises the question of whether PLP, or other similar proteolipids, can adopt different conformations in the cell to adapt their structure to distinct environments.

A second feature of MAL-family proteins is that they partition very efficiently into detergent-resistant membrane (DRM) fractions, whereas the bulk of membrane proteins are completely excluded or show minimal affinity [12,12,13,14,15]. DRMs are known to be enriched in specialized cellular membrane subdomains with a condensed structure due to the dense lateral packing of sterols and lipids with saturated acyl chains [16]. In some of the cases, the presence of MAL proteins in this type of domain in the cell has been confirmed by the use of fluorescent probes [14,17,18,19], such as Laurdan (6-lauryl-2-dimethylamino-napthalene) [20], which distinguishes densely packed from loosely packed membranes. Based on the different strands of evidence, it was proposed that at least some MAL proteins organize membrane lipids to create distinct condensed membrane environments [21].

MAL2, which is heavily glycosylated in its second extracellular loop, is the only member of the MAL protein family reported to undergo permanent posttranslational modifications [12].

## 3. MAL-Family Protein Function

### 3.1. The MAL-MAL2-MALL Branch

The *MAL* gene was firstly identified in a search for genes expressed during T cell development [22]. It was later shown to be expressed in polarized epithelial cells and myelin-forming cells (for a recent review on MALL, see [21]). In polarized epithelia, MAL mostly distributes in a subapical endosome compartment, although it is also present at the plasma membrane and in the Golgi region, which are locations where MAL is involved in membrane trafficking (Figure 2A). MAL was identified as an element of the machinery for a route that transports newly synthesized proteins to the apical membrane directly from the Golgi. Exogenous MAL is able to establish a direct route even in cells, such as hepatocytes, that lack this route [21,23], which reinforces its proposed role in apical trafficking. To do this, MAL continuously shuttles in vesicular structures between the apical membrane, endosomes and the Golgi, following an itinerary that might be controlled by sorting signals present at its C-terminal end. The return of MAL from the plasma membrane to the Golgi is exploited for a clathrin-dependent specialized route of apical endocytosis, thereby contributing to the homeostasis of the apical surface. MAL has also been involved in apical retention of membrane proteins, lumen and primary cilium formation, and as receptor for the clostridial epsilon toxin. In myelin-forming cells, MAL regulates the correct localization of components that mediate axon-glia interactions during ensheathment and myelin wrapping. MAL was shown to be responsible for the vesicular transport of Lck to the plasma membrane in T cells (Figure 2B) in a series of articles [13,18,24,25]. Results of a recent study using an antibody that recognized cells expressing MAL (but not those that not do not express this protein) by flow cytometry have cast doubt on this role. This antibody, however, was not validated by biochemical means [26]. In addition, MAL is crucial for exosome biogenesis by T cells, and important for the proper subcomparmentalization of the zone of contact, known as the immunological synapse, between the T cell and an antigen-presenting cell [21].

MAL2 was originally identified by its interaction with TPD52-like proteins [27], a family of proteins frequently overexpressed in cancer [28]. MAL2 distributes in subapical endosome structures [12,29] and was found to be essential for the indirect route of apical traffic, which transports protein cargo from the basolateral to the apical membrane by a process known as transcytosis [12,30]. For this process to occur, MAL2 travels in vesicular carriers towards the basolateral surface to fuse with peripheral endocytic basolateral vesicles (Figure 2C). The fused vesicles then transport the endocytosed cargo to the apical surface with the participation of the formin INF2 [30,31]. In addition to transcytosis, MAL2 has been reported to regulate the delivery of specific cargo from the Golgi to the plasma membrane [32].

The *MALL* gene, also called *BENE*, was independently identified in searches for non-*V_κ_* transcripts from the immunoglobulin *κ* locus [33] and for candidate genes for familial juvenile nephronophthisis [34,35]. The primary structure of MALL contains three consensus sequences of interaction with the scaffolding domain of caveolin-1, a component of flask-shaped invaginations of the plasma membrane called caveolae. In a first characterization, MALL was found to be associated with caveolin-1 and caveolin-2, probably via these sequences [6]. Using distinct methods of cell fixation, endogenous MALL was recently shown to distribute in membranes and nuclear PML bodies. This latter finding was unexpected since PML bodies are membraneless structures that are formed in an aqueous environment [36,37]. These observations suggest that the MALL proteolipid adopts a membrane-tetraspanning or a water-soluble conformation depending on its physical environment—lipidic or aqueous—in the cell [38]. This capacity is consistent with the conformational changes observed in vitro for the myelin proteolipids PLP and PLLP [9,10,11]. The pool of MALL in membranes was reported to collaborate with caveolin-1 in cholesterol homeostasis and/or caveolin-1 transport [6], whereas the pool in PML bodies is probably related to some of the multiple functions assigned to PML bodies [38].

### 3.2. The PLLP-CMTM8 Branch

PLLP was initially isolated from plasma membranes of bovine kidney plasma membrane [39] and was later characterized as a component of synaptic plasma membranes, myelin sheaths, and endocytic vesicles [4,40]. PLLP, which mostly distributes in subapical endosomes, is required for apical endocytosis and endosomal maturation [41,42], and acts in the recycling of apical proteins from sorting endosomes to the apical membrane (Figure 2D). Important cargoes of PLLP-containing vesicles are the transmembrane proteins Crumbs and Notch, which are crucial master regulators of epithelial morphogenesis and differentiation, respectively. Consistent with this role, PLLP was shown to be required for apical endocytosis and epithelial morphogenesis in the gut of zebrafish [41]. PLLP has also been involved in transcytotic transport of intercellular adhesion molecule (ICAM)-1, a transmembrane protein that mediates firm adhesion of leukocytes to epithelial endothelial cells, to the canalicular surface of hepatic cells [42].

CMTM8—formerly known as chemokine-like factor superfamily 8 (CKLFSF8)—is sometimes considered to be a member of the CMTM family, although it is structurally more similar to PLLP (43% amino acid identity) than to the other CMTM proteins [43]. Consistent with this observation, our bioinformatic analysis of the primary structure of the entire MARVEL superfamily places CMTM8 within the MAL family, CMTM8 being a link with the CMTM family [14,21]. The cellular function of CMTM8 has been associated with endocytosis of the EGFR, a receptor with tyrosine kinase activity, whose levels or activity are affected in many types of cancers. GFP-CMTM8, which localizes to the plasma membrane, internalized with the EGFR in the same endocytic structures following EGF stimulation. CMTM8 overexpression did not affect the initial activation of EGFR, but desensitization of EFG-induced signaling occurred rapidly and correlated with increased endocytosis of the receptor. Conversely, CMTM8 knockdown (KD) delayed EGFR endocytosis and extended the period of EGF-induced signaling [44]. Therefore, CMTM8 acts as an attenuator of EGF-induced signaling by facilitating ligand-induced EGFR endocytosis and subsequent desensitization, indicating that the alteration of CMTM8 levels in cancer might affect EGFR signaling.

### 3.3. The MYADM-MYADML2 Branch

*MYADM* was identified by mRNA differential display analysis as a strongly upregulated gene in a multipotent progenitor cell line induced to differentiate into granulocytes and macrophages [45,46]. Exogenous MYADM localizes to the plasma membrane [47], where it coincides with the small Rho-family GTPase Rac in membrane protrusions [14]. MYADM KD cells showed altered membrane condensation, suggesting a role for MYADM in the organization of condensed membranes (Figure 2E). In addition, these cells showed deficient incorporation of Rac to DRMs, and reduced cell spreading and migration. These findings led to the proposal that MYADM is an organizer of condensed membranes at the cell surface for selective recruitment of Rac and other proteins normally residing in this type of specialized membrane [14]. In endothelial cells, MYADM, probably through the modulation of membrane condensation, regulates the expression of ICAM-1 and the activation of ezrin, radixin, and moesin, thereby controlling the endothelial barrier function and inflammatory response [48].

## 4. The *MAL* Gene Family

### 4.1. Chromosome Location, Exon/Intron Organization, and CpG Island Content

The *MAL* and *MALL* genes are located on the long arm of chromosome 2, approximately 15 Mb apart. All the other *MAL*-family genes are situated on different chromosomes: CMTM8 on chromosome 3, *MAL2* on chromosome 8, *PLLP* on chromosome 16, and *MYADML2* and *MYADM* on chromosomes 17 and 19, respectively (Appendix A).

CpG islands are regions of the genome that contain a large number of CpG dinucleotide repeats and a G+C base content greater than 50% [49,50]. In mammalian genomes, CpG islands typically extend for 500–1500 base pairs and are usually located within gene promoter regions, sometimes extending into the first exon and 3′ downstream regions. DNA methylation is one of the most common molecular alterations in human neoplasia. The absence of DNA methylation is associated with active gene transcription, whereas hypermethylation often results in transcriptional blockade [50,51]. The analysis of the presence of CpG islands shows that all *MAL*-family genes, except *MYADM* and *MYADML2*, contain a CpG island at their promoter region and, therefore, are susceptible to regulation by DNA methylation (Appendix A).

### 4.2. Tissue Expression

The Genotype-Tissue Expression (GTEx) portal (gtexportal.org, accessed on 24 February 2023) is a public resource that contains gene expression datasets from multiple normal tissues. A heatmap of the transcript abundance corresponding to the *MAL*-family genes in a large GTEx panel of tissues is presented in Figure 3 (see also Appendix A). Except for *MYADM*, which is ubiquitously expressed, the other genes are not significantly expressed (<3.7 transcripts per million, TPM) in all the tissues analyzed. *MYADML2*, which is detected mainly in skeletal muscle and to a lesser extent in brain and testis, is the most extreme case. All the tissues represented in the panel express at least three *MAL*-family genes with an abundance of >3.7 TPM. The expression of MAL-family proteins in different human tissues has been analyzed by IHC in the cases of MAL and MAL2 [52,53].

### 4.3. Association with Non-Cancerous Diseases

A missense variant of MAL (p.Ala109Asp) was identified as causing a rare, hypomyelinating leukodystrophy [54] similar to Pelizaeus–Merzbacher disease [55]. *MAL2* gene expression was found to be upregulated in the subcutaneous adipose tissue of Huntington’s disease patients [56]. A computational analysis of transcriptomic data showed *MAL2* to be one of the strongest hub genes involved in signaling pathways in diabetes type 2 connected to tuberculosis and rheumatoid arthritis, although the contribution (if any) of *MAL2* to the disease is unknown [57]. Altered *PLLP* expression has been related to a number of neurological diseases (e.g., schizophrenia, depressive disorder, and Alzheimer’s disease), diabetes type 2, cornea degeneration, lung sarcoidosis, and hyperalphalipoproteinemia [40]. A number of transcriptome-wide meta-analyses aimed at identifying genes involved in hypertension detected *MYADM* as a candidate [58,59,60]. *MYADM* has been implicated in atherosclerotic disease and hypertension as a target of miR-182 [61,62,63], which was previously found to modulate human smooth muscle cell differentiation, proliferation, and migration [63]. It is also known to be relevant to asthma severity as a candidate partner of surfactant protein (SP)-A [64], a protein secreted by alveolar type 2 cells that protects the lung by enhancing the clearance of microbes and modulating the inflammatory response [65]. A homozygous deletion affecting *MYADML2* was found in a consanguineous family with an unusual combination of skeletal abnormalities. Since the deletion affects also the *pyrroline-5-carboxylate reductase*-1 gene, which has been linked to recessive cutis laxa (a skin anomaly that gives the appearance of premature ageing and that usually occurs alongside various other disorders) it remains unclear whether the biallelic deletion of *MYADML2* is the cause of the skeletal malformations [66].

## 5. The *MAL* Gene Family in Human Cancer

Dysregulated expression of MAL-family proteins in specific types of cancer is supported by large-scale analysis of gene expression in patient specimens, such as those in TCGA and by the results of other published studies. The first offers a global panorama of transcript levels, the second a more detailed view (transcript levels, promoter methylation, protein levels, protein subcellular localization, in vitro functional assays, experiments with animal models, etc.), and both yield information that is potentially relevant to cancer prognosis. In the next sections, we have combined the two sources of information to provide an up-to-date overview of the *MAL*-family genes and their protein products in cancer, with particular emphasis on their level of expression, regulation, function, and potential use as therapeutic targets and cancer biomarkers.

### 5.1. Expression of MAL-Family Genes in Cancer

There are many reports of alterations of *MAL*-family expression in cancer. Low *MAL* transcript levels were detected in tumors from breast invasive carcinoma (BRCA) [67], esophageal squamous carcinoma (ESCA) [68,69], head and neck squamous cell carcinoma (HNSC) [70,71,72,73,74], stomach adenocarcinoma (STAD) [75,76], colon (COAD) and rectum adenocarcinoma (READ) [77,78,79,80,81,82], lung adenocarcinoma (LUAD) and lung squamous cell carcinoma (LUSC) [83,84], cervical squamous cell carcinoma (CESC) [85,86,87], and bladder urothelial carcinoma (BLCA) [88,89,90]. With regard to the expression of MAL in lymphomas, IHC and flow cytometry analyses showed MAL to be overexpressed in primary mediastinal large B-cell lymphomas [91], although this was not the case in diffuse large B-cell lymphomas and other B-cell derived tumors [91,92,93,94].

*MAL2* is upregulated in multiple human cancers relative to normal tissues and in cancer-derived cell lines, including BRCA [95,96,97], pancreatic adenocarcinoma (PAAD) [98,99,100], HNSC [101], COAD-READ [102], ovarian cancer (OV), STAD [103,104,105], uterine corpus endometrial cancer (UCEC) [103,104], mesothelioma [106] and neuroblastoma [107]. More puzzling are the cases of malignant liver hepatocellular carcinoma (LIHC), cholangiocarcinoma (CHOL), and renal cell carcinoma, in which, MAL2 was shown by IHC to be downregulated relative to benign tumors [108], but upregulated in cholangiocarcinoma in another study [109]. A similar case is that of meningioma, in which *MAL2* was found to be strongly expressed in benign tumors, but silenced in malignant tumors; this is an example of de novo silencing by DNA methylation [110]. However, in an IHC study, MAL2 was detected in both benign and malignant meningiomas [111]. Comparative experiments with reliable antibodies and parallel analysis of *MAL2* mRNA levels are needed to determine the causes of these apparently conflicting results.

*MALL* expression was upregulated in PAAD [38], but reduced in 70–75% of COAD-READ cancers compared with normal tissues [112,113]. More than two-thirds of primary CESC and some cervical cancer cell lines (CaSKi and HeLA) express low levels of *MALL* transcripts, whereas the level of expression in other cell lines (HT-3 and D98-AH2) is similar to that of normal cervical squamous cells [85]. *MALL* is also downregulated in LUSC and LUAD relative to normal tissue [114]. Although increased gene methylation and loss of heterozygosity was associated with low levels of expression of *MALL* in COAD-READ [113], this seems not to be the case in CESC [85].

Less information is available about the expression levels of other *MAL*-family genes. Using xenografts in nude mouse models of human breast cancer and melanoma metastasis, *PLLP* was found specifically upregulated in tumor cells that metastasize to the brain [115]. Using IHC staining, CMTM8 was found to be significantly downmodulated or absent in LIHC, LUSC, COAD, READ, ESCA, STAD, and BLCA compared with adjacent normal tissue [116,117,118], but upregulated in PAAD [119]. *MYADM* expression was found to be downregulated in skin cutaneous melanoma (SCKM) metastasis compared with nevocellular nevus [120].

GEPIA2 (gepia2.cancer-pku.cn, accessed on 24 February 2023) is an interactive web server for gene expression analysis that uses the TCGA and GTEx datasets. It provides, amongst other outputs, comparative gene expression information about tumors and their corresponding normal tissues [121]. We used GEPIA2 to obtain a panoramic perspective of the expression of all the *MAL*-family genes in a list of 21 cancers that included BLCA, BRCA, CESC, COAD, ESCA, HNSC, kidney chromophobe carcinoma (KICH), kidney renal clear cell carcinoma (KIRC), kidney renal papillary cell carcinoma (KIRP), LICH, LUAD, LUSC, OV, PAAD, prostate adenocarcinoma (PRAD), READ, sarcoma (SARC), SCKM, STAD, thymoma (THYM), and UCEC (Figure 4 and Appendix A).

### 5.2. MAL-Family Gene Methylation in Cancer

Reports about promoter methylation of *MAL*-family genes in cancer are available only for *MAL*, *MAL2,* and *MALL*. Hypermethylation of *MAL* was found in BRCA [67,122], CESC [87], COAD-READ [77,78,79,81,82,123,124], ESCA [125], HNSC [70,71,74], Wilm’s tumor [126], LUSC [83,84], STAD [75], and PRAD [127,128]. Hypermethylation of *MAL2* was observed in malignant meningioma [110], and of *MALL* in COAD-READ [113], but not in CESC [85]. Rescue-of-expression experiments consisting of the treatment of representative cancer cell lines with 5-aza-2′-deoxycytidine, which inhibits DNA methylation, combined or not with trichostatin A, which is an inhibitor of DNA deacetylation, provided additional evidence of hypermethylation being the main cause of *MAL* gene expression suppression in distinct cancers [67,70,71,78,80,81,87,122,125,129].

Hypomethylation was found in a specific region of the transcription start site of the *MAL* promoter, whereas an upstream region was methylated in OV [130]. It is interesting to note that, although *MAL* is hypomethylated and is normally expressed in OV and ovarian cancer cell lines, its expression is upregulated in response to treatment of ovarian cancer cell lines with 5-aza-2′-deoxycytidine, indicating that the methylated region at the promoter region regulates *MAL* expression [130]. Hypomethylation was also found in *MAL2* in OV [131].

### 5.3. Amplification, Deletion and Mutation of MAL-Family Genes in Cancer

The cBioPortal (cbioportal.org, accessed on 24 February 2023) web server allows visualization and analysis of large-scale cancer genomics datasets [132,133]. We used cBioportal software (Memorial Sloan Kettering Cancer Center, New York, NY, USA) to address whether *MAL*-family copy number alterations (CNAs) or mutations are frequent in cancer (Figure 5). The frequency of mutations was less than 2% in all the cases examined. *MAL2* was the *MAL*-family gene with the highest percentage of CNAs, being particularly high in OV (22.0%), BRCA (12.5%), LICH (10.5%), and PAAD (8.7%). OV is the type of cancer that accumulates most CNAs in *MAL*-family genes, and UCEC has the most mutations. The amplification of *MAL2* observed in most of the cancers examined is consistent with the presence of the *MAL2* gene on chromosome 8q24 (Appendix A), a region that is frequently amplified in cancer [134]. In aggressive clones of the SH-SY5Y human neuroblastoma cell line, copy number gain of *MAL2* resulted in increased expression of *MAL2* transcripts [107]. The analysis with cBioportal tools indicates that the percentage of *MAL2* CNAs is positively correlated with the level of *MAL2* transcripts in cancer (Appendix A), as was noted previously [131]. The correlation was less strict for the other *MAL*-family genes (Appendix A).

### 5.4. MAL-Family Gene Regulation by Non-Coding RNA

Non-coding mRNAs (ncRNAs) modulate the expression of protein-coding genes. ncRNAs are classified as short (e.g., micro RNA (miRNA)), or long RNA (lncRNA), depending on whether they are shorter or longer than 200 bp, respectively [135]. miRNAs modulate gene expression by binding to the 3′-untranslated region of their target mRNA [136]. Circular (circ) RNAs, which are a subclass of lncRNA structured in a loop with the 3′ and 5′ RNA ends joined covalently, have a negative effect on miRNA activity [137], whereas other lncRNAs positively contribute to miRNA-mediated gene regulation [138]. Several examples of *MAL*-family gene regulation by ncRNA have been described: *MAL2* expression is modulated by miRNA-129 in papillary thyroid carcinoma [139], by miR-802 together with circ_0084904 RNA in cervical cancer [140], and by miRNA-384 and miRNA320a in collaboration with the lncRNA *Metastasis-Associated Lung Adenocarcinoma Transcript 1* (*MALAT1*) ([141], and LINC00460 [142], respectively; *PLLP* expression is regulated by circ-0002538 and miRNA-138-5p in Schwann cells [143]; *MYADM* is targeted by miRNA-182-3p to regulate hypoxia-induced pulmonary hypertension [61,62]; and miRNA-582-5p inhibits *CMTM8* expression, promoting triple-negative breast cancer invasion and metastasis [144].

In conclusion, for *MAL* and probably for other genes in the family, dysregulation in cancer is a consequence of gene methylation, although CNAs and non-coding RNA contribute to varying degrees depending on the gene and the type of tumor.

### 5.5. Role of MAL-Family Proteins in Cancer

Raft membranes have been implicated in a number of critical membrane transport and signaling processes that cancer cells subvert in order to de-differentiate, migrate, and invade other tissues. Therefore, it is not surprising that raft distortion can contribute to cancer progression, for instance, by participating in the formation of new structures, such as invadopodia, that increase the invasive potential of cancer cells or by affecting signaling pathways mediated by raft-associated receptors. At least some of the members of the MAL family of proteins are colocalized with condensed/raft membranes and play a role in their maintenance and function. Consequently, MAL-family proteins could affect cancer progression indirectly, by affecting the degree of raft condensation, or directly by repurposing their function. An example of MAL-family proteins affecting raft condensation is that of MAL in breast cancer cells, in which MAL levels affected the protein composition of membrane rafts (Figure 6A), with consequences for cell motility [67]. Another example is the downmodulation of *MYADM* in melanoma cells [120], which are known to use two different modes of movement during migration in three-dimensional extracellular matrices. Ameboid or rounded movement allows cells to squeeze forward through gaps present in the extracellular matrix in the absence of proteolytic activity, whereas the alternative mode called elongated or mesenchymal mode is driven by extension of lamellipodia and requires proteolytic processing of the extracellular matrix [145]. MYADM silencing in HeLa cells decreases raft membrane condensation and the presence of Rac in raft membranes (Figure 6A), and, consequently, the formation of Rac-dependent lamellipodia and the mesenchymal mode of movement [14]. This implies that the downmodulation of *MYADM* in melanoma cells might favor ameboid over mesenchymal movement during invasion (Figure 6B).

MAL2 expression in breast cancer offers an example of protein function repurposing to favor cancer progression. Although its normal function is to transport proteins to the apical surface, overexpressed MAL2 in breast cancer cells promoted the endocytosis of tumor antigens via direct interaction with antigen-loaded MHC-I molecules (Figure 6C). In this way, malignant cells impede tumor antigen presentation since they are depleted of antigen-bound MHC-I on their surface, and cannot be recognized and escape from being killed by cytotoxic CD8^+^ T lymphocytes, allowing tumor growth [146]. Increased MAL2 levels in breast cancer cell lines favored the formation of signaling complexes with the transmembrane tyrosine kinase receptor HER2 in membrane protrusions (Figure 6D), contributing to the persistence of activated HER2 on the plasma membrane [147]. In pancreatic cancer cell lines, overexpressed MAL2 associated with the scaffolding protein IQGAP1 that, in turn, activated MAPK ERK1/2 signaling [148] (Figure 6E). Activation of MAPK signaling by excess of MAL2 in non-small lung cancer cell lines activated mTORC1, promoting cell proliferation [149].

MALL overexpression promotes the formation of large solid aggregates that produce cytokinesis failure, leading to cells with aberrant chromosome content [38]. Since MALL is highly upregulated in pancreatic cancer [38], it could produce aneuploidy, which is a hallmark of cancer. Therefore, excess of MALL in some cancers could contribute to malignancy by inducing chromosome instability (Figure 6F).

*MYADML2* overexpression has recently been identified in a CRISPR activation-based screening for LIHC growth and metastasis in nude mice. *MYADML2* overexpression promoted, and its silencing suppressed, cell proliferation and invasion in human hepatic cell lines [150].

### 5.6. The MAL-Protein Family as Tumor Suppressors

Tumor-suppressor genes prevent cancer by impeding the dysregulation of pathways necessary for controlled growth and homeostasis. They are downregulated in cancer, provoking an enhancement of cellular functions that favor cancer progression, for instance proliferation and migration, or a reduction of those that prevent it, such as DNA repair, apoptosis, or elimination of the malignant cells by the immune system [151]. The analysis of the effect on these functions of the exogenous expression of the tumor-suppressor candidate in cell lines that lack the endogenous protein is a useful approach to examine the potential suppressor role played by downmodulated genes. In addition, the comparison of the volume of the tumor produced in nude mice by the control and the modified cancer cell lines yields in vivo information about the tumor-suppressor activity of the gene. Thus, exogenously-expressed MAL in breast [67], esophageal [129], cervical [87], colorectal [113], lung [84], and head and neck [71] cancer cell lines reduced cell migration and invasion, or increased apoptosis, and reduced tumor size in nude mice [71,84,129]. Similar effects were observed for exogenous MAL2 in prostate [152], pancreatic [99], lung [149], and liver [108] cancer cells; for exogenous MALL in colorectal cancer cells [113]; and for CMTM8 in liver [153], pancreatic [119], and urothelial [116] cancer cells. These observations suggest that at least some of the *MAL*-family genes downmodulated in tumors might have a tumor-suppressor function in specific organs and tissues.

### 5.7. MAL-Family Proteins as Potential Therapeutic Targets in Cancer

Genes that contribute to tumor progression facilitate processes that lead to cancer growth and dissemination. For this role, these genes are usually expressed in an upregulated manner in the tumor cells and their high level of expression is usually associated with poor outcome. Their silencing produces similar effects to those of exogenous expression of tumor suppressors. Identifying these genes is of biomedical importance because they are potentially useful as pharmacological targets in cancer.

*MAL2* gene upregulation has been observed in cancers affecting many organs, likely due to CNA (Figure 5 and Appendix A). MAL2 silencing in non-small cell lung cancer cells decreased cell proliferation in vitro, whereas MAL2 overexpression had the opposite effect, and increased tumor size in xenografts in nude mouse models [149]. However, overexpression of MAL2 in a colorectal cell line inhibited cell proliferation and invasion and suppressed tumorigenesis in xenograft in the same models [154], indicating that the role of MAL2 in cancer progression might be tissue-dependent. An effect of MAL2 silencing on cell proliferation, similar to that in lung cancer cell lines, was observed in breast [96], ovarian [131], pancreatic [148] and prostate [155] cancer cell lines, in which migration and invasion and the epithelial-to-mesenchymal transition were decreased. MAL2 silencing also affected tumor size in nude mice xenografts of pancreatic cancer cells [155]. However, no effect on cell proliferation was observed in a study in which MAL2 was either overexpressed or silenced in mouse and human breast cancer cell lines [146]. The role of MAL2 in these processes in breast cancer should therefore be carefully re-evaluated. It is interesting, though puzzling, that the same study shows that tumor growth in immunocompromised nude mice was not significantly different among control cancer cells or the same cells with MAL2 overexpressed or downmodulated, whereas MAL2-overexpressing cells and MAL2-KD cells produced tumors with significantly higher and lower, respectively, volume and weight compared to those of control cells when the same assay was performed in immunocompetent mice [146].

Silencing CMTM8 suppressed, and its overexpression enhanced, the migratory and invasive capacity of human pancreatic cancer cell lines [119]. Consistent with a role of CMTM8 in cancer progression, CMTM8-depleted human pancreatic cancer PANC-1 cells reduced the formation of metastatic lesions in the lung of nude mice relative to those produced by the parental cells [119]. Using human breast cancer and melanoma cell lines inoculated in nude mice, it was found that *PLLP* upregulates in cancer cells metastasized to the brain, raising the possibility that PLLP is involved in the colonization of the brain by these types of cancers [115].

In conclusion, except in colorectal cancer in which MAL2 overexpression is reported to have a protective role in xenograft nude mice models, in this type of model MAL2 appears to favor cancer progression in lung, breast, ovarian, and pancreatic cancer; CMTM8 contributes to the development of metastasis in pancreatic cancer; and PLLP might be involved in the establishment of breast and melanoma metastasis in the brain. Therefore, these MAL-family proteins constitute potential targets for developing therapeutic drugs to alter cancer progression.

### 5.8. MAL-Family Genes as Prognostic Cancer Biomarkers: Analysis of Large-Scale Cancer Datasets

Kaplan–Meier plots portray the association between the expression of a given gene and the probability of patient survival. To make the plots, patients are classified into high-or low-expression groups; the correlation between the two groups and patient survival is examined using clinical data. Once a significant correlation has been found, the gene can be used as a prognostic biomarker in conjunction with other established biomarkers to estimate the probability of survival of other patients. We used the Kaplan–Meier plotter (kmplot.com, accessed on 24 February 2023) and Human Protein Atlas (HPA, proteinatlas.org, accessed on 24 February 2023) plots to assess the correlation between the expression level of the *MAL*-family genes and the probability of overall survival for the 21 cancer types whose transcript levels were analyzed in Figure 4. BLCA, BRCA, HNSC, KICH, KIRC, KIRP, LICH, LUAD, LUSC, OV, PAAD, STAD, THYM, and UCEC yielded a correlation with associated values of *p* ≤ 0.01 between the expression of at least one *MAL*-family gene and the probability of survival, whereas there was no significant correlation or *p* > 0.01 result in CESC, COAD, ESCA, PRAD, READ, SARC, and SCKM (Figure 7 and Appendix A).

Focusing only on the highly statistically significant cases (*p* ≤ 0.001), the correlation with *MAL2* expression in BRCA, which occupied the first position in the list of new cancer cases in 2020 with 12.5% of the cases, is of particular note. In fact, *MAL2* was the fourth highest HPA-ranked gene with the highest significance associated with an unfavorable prognosis in breast cancer. Other stronger correlations were that of *MAL* (LUAD) and *MYADM* (LUSC) in lung cancer, which was the second most highly ranked in the 2020 ranking, with 12.5% of the new cancer cases, and that of *MAL2* and *MALL* in PAAD, which was ranked twelfth, with 2.7% of the new cases. *MAL* (LUAD, THYM, and UCEC), *MAL2* (BRCA, PAAD, and UCEC), and *PLLP* (KICH, KIRC, and UCEC) present strong correlations (*p* ≤ 0.001) in three types of cancer, and *MALL* (KIRC and PAAD), *CMTM8* (KIRC and KIRP), and *MYADM* (LUSC and THYM) do so in two cancer types. Due to its very low level of expression (<1 TPM) in normal and cancerous tissue (Figure 4), *MYADML2* was considered non-evaluable.

The level of transcripts in the tumor samples relative to those in adjacent normal tissue is indicated in Figure 7 in the cases of strong correlation (*p* < 0.001). The greatest difference was that of *MAL2* in UCEC, followed by that of *MAL* in THYM, *MALL* in PAAD, and of *MAL2* in PAAD and BRCA. Appendix A compiles the data of all the cancers and genes analyzed in this study.

### 5.9. MAL-Family Genes as Prognostic Cancer Biomarkers: Analysis of the Literature

Unfortunately, many of the publications describing potential prognostic uses of *MAL*-family genes in cancer rely on the same public large-scale datasets used in our Kaplan-Meier survival analysis, which makes those publications inappropriate for confirming the results shown in Figure 7. Table 1 summarizes correlations reported between overall survival and MAL-family protein or gene expression or methylation of *MAL*-family genes using data other than large-scale public datasets, and which were obtained by IHC, gene methylation studies or RT-qPCR (Table 1). In classic Hodgkin lymphomas of the nodular sclerosis type and OV [156], IHC analysis indicated that a high level of expression of MAL is correlated with a poor outcome [157]. Less significant correlations were observed for *MAL* hypermethylation in COAD-READ [124] and STAD [75,76]. However, in the case of STAD, one study showed that *MAL* hypermethylation, which was accompanied by *MAL* mRNA downmodulation [75], has a favorable prognosis, while *MAL* mRNA downmodulation correlated with an unfavorable prognosis in another study [76]. Consistent with the analysis illustrated in Figure 7, the use of IHC confirmed that a high level of MAL expression is associated with unfavorable outcome in UCEC [152]. With regard to the other MAL-family members, IHC analysis confirmed that high levels of MAL2 [99] and MYADM [117] are associated with poor outcome in PAAD and STAD, respectively. MYADML2 overexpression was associated with poor outcome in LIHC [150]. Although no significant correlations were found in the analysis of the large-scale RNA-seq datasets, IHC indicated that high levels of expression of MAL2 [102] and low levels of expression of MALL [113] are associated with unfavorable prognosis in the cases of COAD-READ and COAD, respectively.

In addition to the potential use of MAL-family proteins as potential biomarkers for cancer prognosis, MAL-family protein detection could contribute to cancer patient diagnosis by helping to identify the specific type of cancer. For instance, *MAL* was found to be expressed in mediastinal but not peripheral diffuse lymphomas. IHC analysis confirmed the selectivity of MAL expression, revealing that approximately 64% of primary mediastinal tumors express MAL [92,93]. Another example is that of KICH, which has morphological features that often overlap with those of oncocytoma. The distinction between these two tumors is clinically important, as KICH is malignant and can potentially be aggressive, whereas oncocytoma is a benign neoplasm. *MAL2* mRNA and protein were found to be expressed in KICH but not in most oncocytoma [158], making MAL2 a promising biomarker to discriminate these two types of cancer.

The analysis of *MAL* gene in biopsies could also enhance identification of patients with aggressive prostate cancer whose clinical data would otherwise lead them to be considered being of only low or intermediate risk [127,128]. *MAL* expression could also be used to identify a group of bladder cancer patients with unfavorable prognosis [88], and to provide information about progression from pre-invasive leukoplakia lesions to invasive oral squamous carcinoma [159]. With regard to oncovirus-related cancers, *MAL* promoter methylation and downregulation [87] correlated with the development of CESC, which is caused by persistent infection with high-risk human papillomavirus, suggests that *MAL* could be used as a diagnostic biomarker to reduce errors and misjudgment in the analysis of biopsies and scrapings [160] and to predict the severity of the cancer [161,162,163,164]. Another study suggests that *MAL* methylation could also be used as a biomarker in nasopharyngeal carcinomas [165], which is a tumor associated with Epstein–Barr virus, and in epithelial ovarian cancers positive for high-risk human papillomavirus [166]. In the case of Merkel cell carcinoma, which is a rare, aggressive form of skin cancer mainly related to Merkel cell infection by Merkel cell polyomavirus, high levels of MAL expression correlate with favorable outcomes [167].

The development of noninvasive molecular tools for cancer cell analysis is becoming an important advance in cancer cell identification and characterization. Detection of circulating tumor cells, cancer biomarkers, cell-free DNA, DNA methylation, and specific mRNA or miRNA levels in biofluids (liquid biopsies) are used as non-invasive detection tests for cancer screening [168]. Examples of the potential use of the MAL-family members as cancer biomarkers in these types of test are *MAL* methylation assays in blood-derived samples from patients with COAD-READ [169], BRCA [170,171], and CESC [172], and in urine and feces from BLCA [89,90] and COAD-READ [173] patients, respectively; and analysis of *MAL2* transcript levels in blood from gynecological and metastatic breast cancer patients [174,175].

In summary, incorporation of antibodies to MAL-family members into current antibody panels used in routine clinical practice to detect cancer biomarkers in biopsies and surgical specimens, or the use of the approach with liquid biopsies, could be useful means of obtaining prognostic/diagnostic information in many types of cancer.

### 5.10. MAL-Family Expression as Biomarkers to Predict Response to Cancer Chemotherapy

Predictive biomarkers anticipate the possible benefit from a specific treatment, helping professionals to select a particular treatment from a range of alternatives. MAL-family proteins might not only be relevant as prognostic biomarkers, but also could serve as predictive biomarkers. For instance, *MAL* was the most strongly upregulated gene in an α-interferon resistant variant cell clone derived from a sensitive cutaneous T-cell lymphoma cell line, and is expressed in 80% of cutaneous T-cell lymphoma patients showing a slow response to treatment with α-interferon and/or photochemotherapy. It is of note that the level of expression of MAL in the tumor cells was correlated with clinical data concerning time of remission, whereby MAL-positive tumors responded to the treatment significantly more slowly than did the MAL-negative tumors [176]. Another example is that of MAL in high-grade serous carcinoma, which is responsible for the greatest number of cases and the high fatality rate of OV. *MAL* was among the most highly expressed genes in serous OV from short-term survivors (<3 years) relative to those of long-term survivors (>7 years), all of whom were treated with platinum-based therapy [177]. *MAL* transcripts were found significantly overexpressed in ovarian cancer cell lines resistant to traditional platinum-based and other chemotherapeutics compared with sensitive ones [130,178] and in the tumor cells of platinum-resistant compared to platinum-sensitive patients that received platinum-based chemotherapy [178]. Therefore, it was proposed that *MAL* serves to predict the response to platinum-based drugs and as a target for the development of novel therapies aimed at enhancing the sensitivity of OV to these drugs. Another example concerning *MAL2* relates to the sensitivity of pancreatic cancer cell lines to different chemotherapeutic agents (gemcitabine, 5-fluoruracil and cisplatin). The level of *MAL2* transcripts is inversely correlated with resistance to these drugs, suggesting that the analysis of *MAL2* levels in PAAD could be used as an indicator of the response to chemotherapy in pancreatic cancer [179]. *MYADM* was identified in a search for genes upregulated in colon cancer cell lines sensitive to the drug MS-275, N-(2-aminophenyl)-4-[N-(pyridine-3yl-methoxy-carbonyl) aminomethyl] benzamide, which is a second-generation histone deacetylase inhibitor. Therefore, MYADM is a potential biomarker predicting the response to MS-275, and probably other histone deacetylase inhibitors, in COAD.

Although no significant correlation between MAL protein expression and overall survival was found by IHC in a complete cohort of BRCA patients, the Kaplan–Meier analysis of the group of patients who did not receive chemotherapy showed that survival may have been favored by adjuvant therapy (*p* = 0.003) in the case of low-level MAL expression [67]. The receiver operating characteristic curves (ROC) plotter (rocplot.org, accessed on 24 February 2023) generates ROC graphs useful to link gene expression and response to therapy using patient transcriptome data from four types of cancer [180]. Confirming the potential use of MAL as a predictive biomarker in BRCA [67], by using the ROC plotter we have found that *MAL* is a biomarker with potential clinical utility to predict the response of breast cancer patients to anthracycline and taxane (Appendix A), which are often used in adjuvant chemotherapy for early breast cancer [181,182]. However, unlike a previous proposal [178], it does not predict any effect of *MAL* expression in the response to platinum-based chemotherapy of ovarian cancer patients.

### 5.11. MAL-Family Proteins and Antitumor Drug Development

Bile acid transporter-targeting is an effective strategy for drug delivery to hepatocytes and enterocytes [183]. The finding that bile acid transporters, such as the apical sodium bile acid transporter (ASBT), are expressed in tumor cells raises the possibility of using bile acid-modified anticancer drugs to treat tumors. For instance, the uptake of polystyrene nanoparticles conjugated with glycocholic acid has been successfully exploited to target human breast cancer SK-BR-3 cells, which naturally express ASBT. However, given the small cavity size of ASBT, the uptake of nanoparticles does not take place by transporter pumping but by internalization together with ASBT in a MAL2-dependent manner [184]. Since MAL2 is overexpressed in breast cancer, breast tumors might possibly be susceptible to treatment with ASBT-targeting nanomedicines.

### 5.12. Tools for MAL-Family Proteins as Cancer Biomarkers

Gene methylation/expression analyses are not routine practices in most medical pathology departments, where histochemistry and IHC are the gold standards for cancer cell characterization. Therefore, a more appropriate approach would be to carry out IHC analysis of protein expression with highly specific antibodies.

As is the case for all proteins, the use of MAL-family proteins as cancer biomarkers requires well-characterized, validated antibodies. Validation criteria should include as many of the following tests as possible: (i) presence and absence of signal by immunoblotting in extracts from cells known to be positive and negative, respectively, for the expression of the endogenous protein in the event that the protein is not ubiquitously expressed, (ii) recognition of exogenously-expressed protein by immunoblot or other techniques, (iii) loss of the signal upon protein silencing (gene KD or, better, KO) in cells positive for the endogenous protein, and (iv) loss of signal by competition with an excess of the immunogen used (peptide or purified protein) to generate the antibody. Unfortunately, this information is poorly covered in scientific publications and commercial datasheets. In Table 2, we have compiled information about the antibodies to MAL-family proteins described in the scientific literature, including the reference in the case of commercial antibodies, the criteria used for their validation, and the techniques for which these antibodies are purported to be useful.

The use of strictly validated antibodies to MAL-family proteins would help avoid ostensibly contradictory results, which often arise from studies in which the expression of a biomarker in cancer samples is examined by different techniques (for instance, RNA-seq and IHC) or using different sources of primary antibodies. The conflicting results regarding MAL2 expression in benign and malignant meningiomas [110,111] are an example of this. An IHC study indicated that MAL2 protein expression is downregulated in LIHC and CHOL compared with the expression in benign adjacent tissue [108], even though chromosome 8q24 amplification suggested that *MAL2* could be overexpressed in these two types of tumor (Figure 5). Consistent with the latter, GEPIA2 analysis indicated *MAL2* upregulation in LIHC and CHOL and, consistent with this result, *MAL2* transcripts were found upregulated in various CHOL-derived cell lines in another study [109]. Another controversial example is that of CMTM8 in COAD in which, consistent with GEPIA2 analysis (Figure 4 and Appendix A), one IHC study found a high level of expression in tumors compared with normal tissue [185], whereas the converse relationship was observed in another IHC study that used a different primary antibody [118].

**Table 2 cancers-15-02801-t002:** Antibodies used in scientific publications.

Protein	Antibody	Source	Address	Validation ^1–5^	Applications ^6^	References
MAL	Mouse 6D9 mAb	MA Alonso	Centro de Biología Molecular, Madrid, Spain	Endog, Exog, +/−, KD/KO, Compet	WB, IHC, IP, FACS	[18,24,25,53,67,69,78,91,92,93,94,129,156,157,176,186,187,188,189,190,191]
	Unknown	Unknown	Unknown	Unknown	WB, IHC	[152]
	Goat polycl MAL (H-70)	Santa Cruz Biotech	scbt.com, discontinued product	Unknown	WB, IF	
	Rabbit polycl	Santa Cruz Biotech	scbt.com, discontinued product	Unknown	IHC	[74]
	Mouse mAb MT3	W Kasinrek	Chiang Mai University, Thailand	Expression cloning; KO	IF, FACS	[26]
MAL2	Mouse 9D1 mAb	MA Alonso	Centro de Biología Molecular, Madrid, Spain	Endog, Exog, +/− KD/KO	WB, IHC, IF, IP	[12,29,30,31,52,146,158,191]
	Rabbit polycl ab217919	Abcam	abcam.com, accessed on 24 February 2023	Unknown	IHC	[111]
	Rabbit polycl ab75347	Abcam	abcam.com, accessed on 24 February 2023	Exog, KD	WB, IHC, IF	[102,108,131,148]
	Rabbit polycl bs-7175R	Bioss Antibodies	biossusa.com, accessed on 24 february 2023	Exog, KD	IWB, IF	[147,149]
	Rabbit polycl	JA Byrne	The Children’s Hospital, Westmead, Australia	Exog	WB, IF, IHC	[95,105,192]
	Rabbit polycl sc-87994	Santa Cruz Biotech	scbt.com, discontinued product	Unknown	IHC	[99]
	Unknown	Unknown	Unknown	Unknown	WB, IHC	[155]
MALL	Mouse 2G8 mAb	MA Alonso	Centro de Biología Molecular, Madrid, Spain	Exog, KD/KO	WB, IHC, IF, IP	[38]
	Rabbit polycl	Abgent	Abgent.com, accessed on 24 February 2023. Product not found in the catalog	Unknown	IHC	[113]
PLLP	Rabbit polycl	J Millán	Centro de Biología Molecular, Madrid, Spain	Exog, KD/KO	WB, IHC, IF, IP	[41,42]
	Rabbit polycl	HW Müller	Heinrich-Heine Univ. Düsseldorf, Germany	Exog	WB, IHC	[15,193,194]
	Rabbit polycl	VS Sapirtstein	Medical College of Pennsylvania, Philadelphia, PA	Exog	WB, IHC	[4,195,196,197,198,199]
CMTM8	Rabbit polycl	Y Wang	Peking Univ. Health Science Center, Beijing, China	Exog, KD, Compet	WB, IHC	[116,118,153,200]
	Unknown	Unknown	Unknown	Unknown	WB, IHC, IP	[119]
	Rabbit polycl 15039-1-AP	Proteintech	ptglab.com, accessed on 24 February 2023	Unknown	IHC	[117,185]
MYADM	Mouse 2B12 mAb	MA Alonso	Centro de Biología Molecular, Madrid, Spain	Exog, KD/KO	WB, IP	[14,48]
MYADML2	Bs-19119R7	Bioss	Biosusa.com, accessed on 24 February 2023. Product not found in the catalog	Endog	WB, IHC	[150]

^1^ Endog: recognition of the endogenous protein by WB. ^2^ Exog: recognition of the exogenous protein by WB. ^3^ +/−: presence or absence of signal in cell lines positive or negative, respectively, for the expression of the endogenous protein by WB. ^4^ KD or KO: loss of endogenous signal in KD or KO cells. ^5^ Compet: loss of WB or IF signal by competition with the peptide/protein used as immunogen. ^6^ Reported applications: WB, Western blot; IHC, immunohistochemistry; IF, immunofluorescence; IP, immunoprecipitation; FACS, fluorescence activated cell sorter.

Analysis of MAL-family protein expression by IHC could also be useful for identifying cancerous or precancerous lesions through a simple analysis of stained specimens in cases in which there is a marked difference between the expression in the tumor and the normal tissue. For instance, *MAL* expression is reduced by more than 1000-fold in ESCA relative to normal adjacent tissue (Figure 4 and Appendix A). Although *MAL* expression is not prognostic in ESCA, loss of MAL protein expression was detected in incipient esophageal neoplasms in rat models and also in mild esophageal dysplasia in humans [69]. Given the very high level of expression of *MAL* in normal esophageal mucosa and its dramatic downregulation in ESCA, monitoring MAL expression by IHC could be useful for characterizing incipient neoplasms in esophageal biopsies. Other examples of dramatic differences in gene expression are the downmodulation of *MAL* in HNSC (250-fold), and its upregulation in OV (30-fold) and THYM (70-fold); *MAL2* upregulation in BRCA (7-fold), CESC (485-fold), COAD (28-fold), OV (750-fold), PAAD (10-fold), READ (almost 90-fold), and UCEC (350-fold); *MALL* upregulation in STAD (almost 10-fold); *PLLP* downmodulation in SARC (10-fold) and SCKM (44-fold downmodulated); and *MYADM* upregulation in PAAD (10-fold) (Figure 4 and Appendix A). If the difference in the abundance of transcripts between normal tissue and tumors is concordant at the protein level, IHC analysis with validated antibodies could be useful for characterizing these types of tumors. Detection of alterations of the expression of MAL-family proteins in liquid biopsies could also serve as a tool for screening in these cases.

In summary, whereas some of the antibodies to MAL-family proteins have been fully validated, others require stringent validation tests to avoid false results.

## 6. Conclusions

Our analysis of large-scale genomic datasets indicates that the expression level of at least one *MAL*-family gene member is strongly correlated with prognosis in nine types of cancer that, according to the World Cancer Research Fund (wcrf.org/cancer-trends/worldwide-cancer-data, accessed on 24 February 2023), cover approximately one-third of the new human cancer cases in 2020. These include BRCA (12.5%), renal cancer (KICH, KIRC, and KIRP; 2.5%), lung (LUAD and LUSC; 12.5%), PAAD (2.7%), THYM (<0.1%), and UCEC (2.3%). In four of the cancers analyzed (KIRC, PAAD, THYM, and UCEC), expression of two or more *MAL*-family genes was closely associated with patient outcome. The information means that MAL-family members are candidates worthy of consideration as cancer biomarkers. Considerable progress has been made towards understanding the function of some of the MAL-family proteins, but there is still a long way to go before their function in normal cells and how it is exploited by tumor cells is completely understood. This is especially evident in the cases of MALL, PLLP, and MYADM, whose function in normal and tumor cells needs further investigation using established cell model systems, and that in the case of MYADML2, which is yet to be examined. The in vivo cancer-related work performed so far on MAL-family proteins has involved inoculating human cancer cell lines in nude mice. Since these mice are immunocompromised and do not reproduce the tumor microenvironment, the inoculated cells metastasize early on, making this model inappropriate for studying cancer progression. To advance our knowledge of the role of MAL-family proteins in vivo, the use of mouse models with genetically modified *MAL*-family genes in specific tissues will be of great help in developing de novo tumors in a natural immune microenvironment, thereby mimicking the process in humans, to allow spontaneous progression toward metastatic disease. The use of this type of mouse will clarify how MAL-family proteins contribute to cancer, and it is hoped that it may serve for the development of specific therapeutic drugs.

Chimeric antigen receptor (CAR)-T cell therapy is a type of novel cancer immunotherapy that uses a patient’s own T cells. CAR-T cells are T cells genetically engineered in the laboratory to express a CAR that recognizes a specific target antigen on the surface of tumor cells. After in vitro expansion, CAR-T cells are reinfused into the same patient to kill tumor cells. CARs are assembled with an antigen-binding domain, consisting of the variable heavy and light chains of a mAb specific to the antigen, fused to a hinge region, a transmembrane domain, and one or more cytoplasmic signaling modules [201,202]. The differential expression of the antigen between the tumor and the normal tissue and the existence of specific mAbs against the antigen are important factors in CAR design. MAL-family proteins overexpressed in tumor cells —for example, MAL2 in BRCA, PAAD, and UCEC, and MALL in PAAD—could serve as putative target antigens for CAR-T cell therapy.

Genomic mutations in tumor cells within the sequence coding for surface proteins can generate protein variants, known as neoantigens, that are completely unfamiliar to the immune system. When the alteration occurs in the promoter region, the genes may be expressed in the wrong type of cell or tissue, or at the wrong time, giving rise to tumor-associated proteins [203,204]. Neoantigens and tumor-associated proteins can serve as targets for vaccine-based cancer immunotherapy, the former being the better choice [205]. The *MAL*-family genes are subjected to CNA, mutations, and epigenetic modifications to varying degrees. Given that most MAL-family proteins are expressed to some extent on the plasma membrane, it is plausible that they can give rise to neoantigens or tumor-associated proteins. This raises the possibility of using MAL-family proteins as putative targets for cancer vaccines and for CAR-T cells immunotherapy.

The possible routine use of MAL-family proteins as cancer biomarkers in medical pathology units requires IHC analysis of a larger number of cancer samples with validated antibodies. Stringent validation in the case of existing antibodies, and the generation and validation of new ones are needed to extend the use of MAL-family proteins as cancer biomarkers. In conclusion, MAL-family proteins are potential targets of therapeutic drugs and their use as cancer biomarkers can provide useful prognostic and diagnostic information in human cancer.

## Figures and Tables

**Figure 1 cancers-15-02801-f001:**
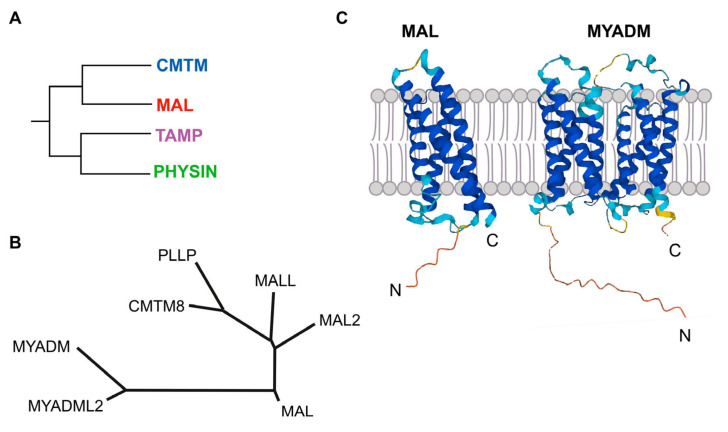
The MAL protein family. (**A**) Schematics of the organization of the MARVEL superfamily of proteins in distinct families, one of which is the MAL family. (**B**) Tree of the MAL-family of proteins. After sequence alignment with ClustalW, the family tree was constructed by the neighborhood-joining method using MEGA 11 software (Pennsylvania State University, University Park, PA, USA). The protein accession numbers are: MAL (NP_002362.1), MAL2 (NP_443118.1), MALL (NP_005425.1), PLLP (NP_057077.1), CMTM8 (NP_849199.2), MYADM (NP_612382.1), MYADML2 (NP_001138585.2). (**C**) Predicted structure of MAL and MYADM according to Alphafold.

**Figure 2 cancers-15-02801-f002:**
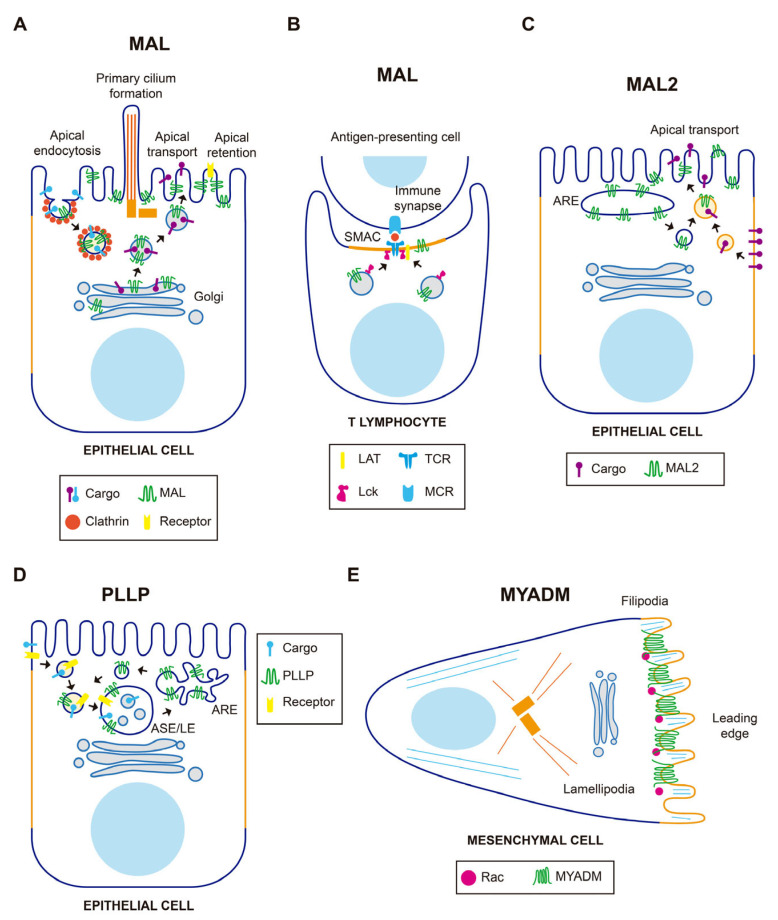
Schematics of the function of MAL-family proteins in normal cells. (**A**) In epithelial cells, MAL is involved in polarized transport to the apical surface directly from the Golgi, apical endocytosis, retention of membrane receptors, and primary cilium formation. (**B**) In T cells, MAL directs the traffic of Lck to the immunological synapse and is important for the maintenance of the proper architecture of the supramolecular activation complex (SMAC). (**C**) MAL2 directs basolateral-to-apical transcytosis of membrane protein cargo. (**D**) PLLP is involved in the recycling of receptors from late endosomes to the plasma membrane. (**E**) MYADM accumulates at the leading edge of migrating fibroblasts to recruit the Rac GTPase. ARE, apical recycling endosome; ASE/LE, apical sorting endosome/late endosome.

**Figure 3 cancers-15-02801-f003:**
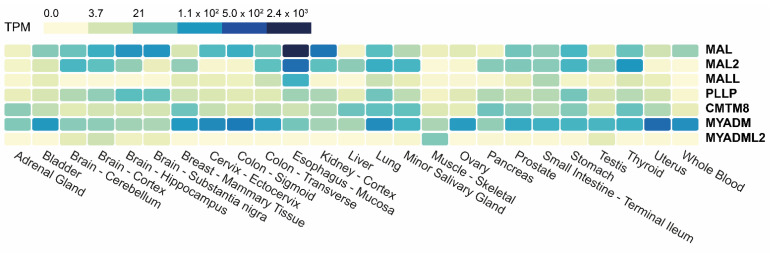
Heatmap of the expression of *MAL*-family genes in different human tissues. The data used were obtained from GTEx. The color scale indicates the number of TPM.

**Figure 4 cancers-15-02801-f004:**
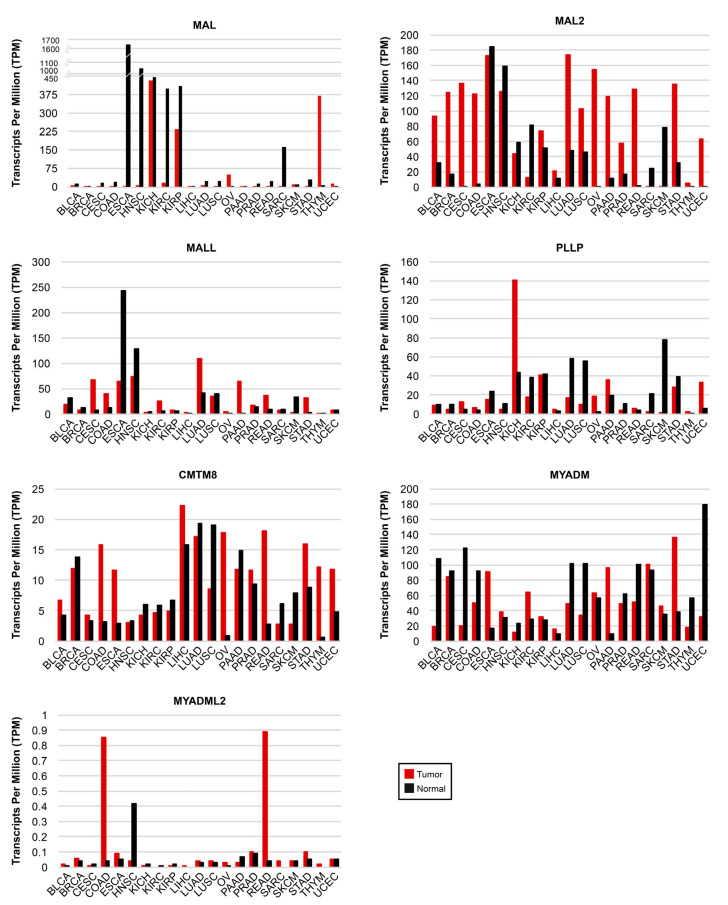
Expression of *MAL*-family genes in human cancers. The expression of each *MAL*-family gene in the indicated type of cancer is compared with that in adjacent normal tissue. GEPIA2 was used for the analysis of RNA-seq data from TGCA and GTEx datasets. Note the use of distinct vertical scales in the panels. BLCA, bladder urothelial carcinoma; BRCA, breast invasive carcinoma; CESC, cervical squamous cell carcinoma; COAD, colon adenocarcinoma; ESCA, esophageal squamous cell carcinoma; HNSC, head and neck squamous cell carcinoma; KICH, kidney chromophobe carcinoma; KIRC, kidney renal clear cell carcinoma; KIRP, kidney renal papillary cell carcinoma; LICH, liver hepatocellular carcinoma; LUAD, lung adenocarcinoma; LUSC, lung squamous cell carcinoma; OV, ovarian serous cystadenocarcinoma; PAAD, pancreatic adenocarcinoma; PRAD, prostate adenocarcinoma, READ, rectum adenocarcinoma; SARC, sarcoma; SCKM, skin cutaneous melanoma; STAD, stomach adenocarcinoma; THYM, thymoma; UCEC, uterine corpus endometrial carcinoma.

**Figure 5 cancers-15-02801-f005:**
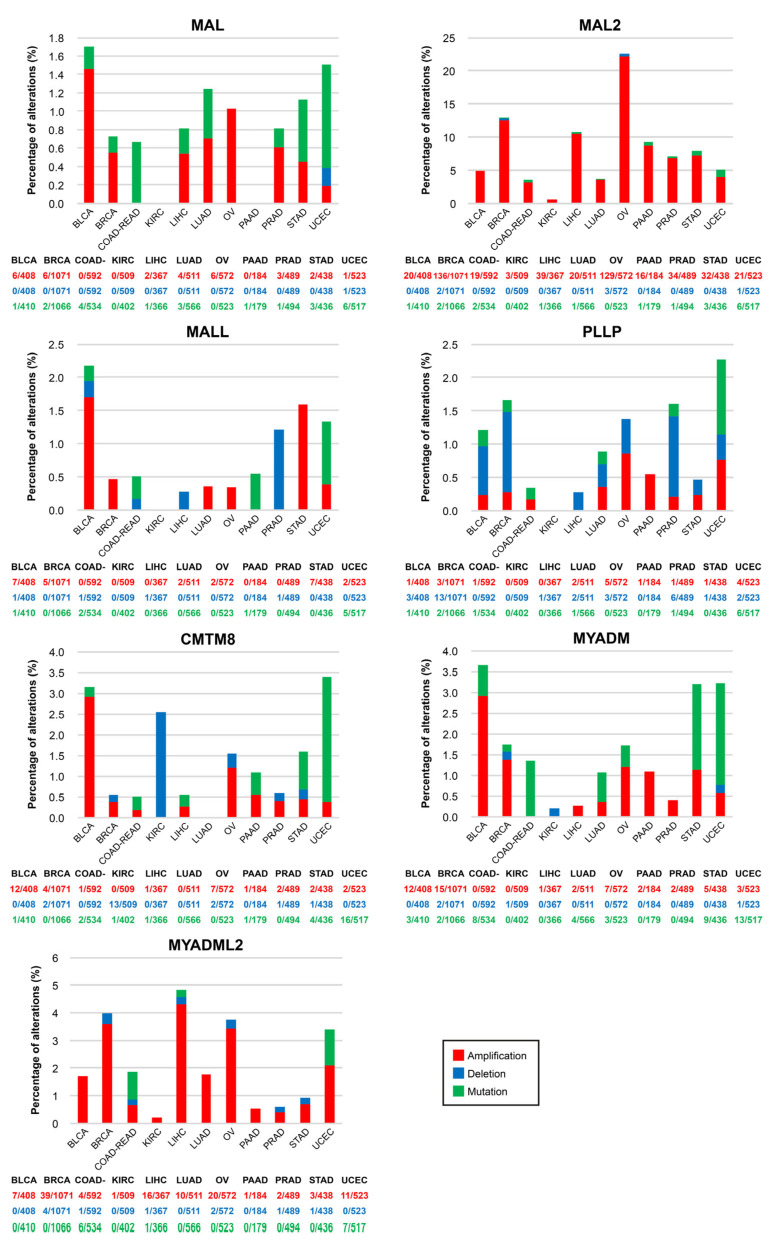
Mutations and copy number alterations of *MAL*-family genes in human cancers. Percentage of tumor samples with mutations, deletions, or amplification of *MAL*-family genes in the indicated cancers. The histograms were generated with cBioportal using TCGA PanCancer Atlas datasets. Note the use of distinct vertical scales in the panels. BLCA, bladder urothelial carcinoma; BRCA, breast invasive carcinoma; COAD-READ, colon-rectum adenocarcinoma; KIRC, kidney renal clear cell carcinoma; LICH, liver hepatocellular carcinoma; LUAD, lung adenocarcinoma; OV, ovarian serous cystadenocarcinoma; PAAD, pancreatic adenocarcinoma; PRAD, prostate adenocarcinoma; STAD, stomach adenocarcinoma; UCEC, uterine corpus endometrial carcinoma.

**Figure 6 cancers-15-02801-f006:**
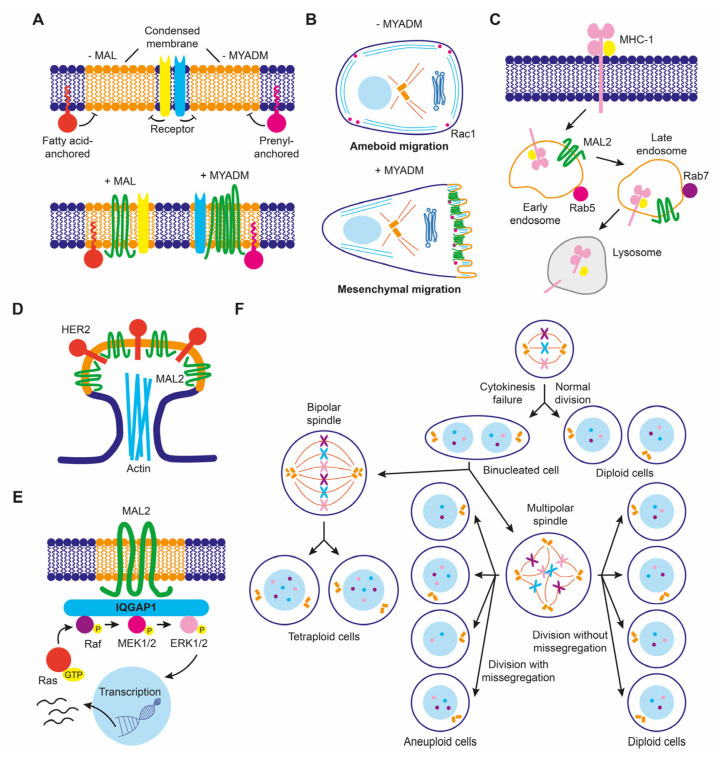
Schematics of some proposed functions of MAL-family proteins in cancer cells. (**A**) MAL and MYADM levels control membrane condensation that, in turn, regulates the confinement of proteins anchored to the inner leaflet of the lipid bilayer through covalently linked fatty acids alone (for instance, Lck) or combined with prenylation (for instance, Rac), respectively, and probably also of specific membrane receptors. (**B**) MYADM levels modulate ameboid versus mesenchymal movement by regulating the recruitment of Rac to condensed membranes. (**C**) MAL2 overexpression modulates endocytosis of MHC-I in breast cancer cells. (**D**) MAL2 organizes specialized membranes to recruit HER2 in breast cancer cells. (**E**) MAL2 regulates MAPK ERK1/2 signaling in pancreatic cancer cells. (**F**) Excess of MALL produces cytokinetic defects that lead to aneuploidy. MALL-overexpressing cells may divide normally or undergo cytokinesis failure, giving rise to a binucleated cell (or, if the two nuclei fuse, to a tetraploid cell). Division of this cell can generate diploid, tetraploid, or aneuploid cells depending on the position of the centrosome in metaphase and the occurrence or not of chromosome missegregation.

**Figure 7 cancers-15-02801-f007:**
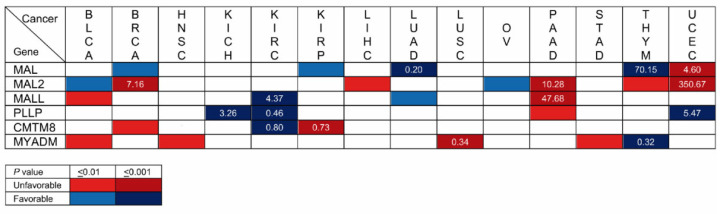
Summary of Kaplan–Meier plot analysis of the correlation between high expression of *MAL*-family genes and overall patient survival. Unfavorable and favorable prognoses are indicated in red and blue, respectively. The intensity of the color indicates the range of the resulting *p* values. The number in the boxes indicates the level of transcripts in the tumor samples relative to that in adjacent normal tissue. BLCA, bladder urothelial carcinoma; BRCA, breast invasive carcinoma; HNSC, head and neck squamous cell carcinoma; KICH, kidney chromophobe carcinoma; KIRC, kidney renal clear cell carcinoma; KIRP, kidney renal papillary cell carcinoma; LICH, liver hepatocellular carcinoma; LUAD, lung adenocarcinoma; LUSC, lung squamous cell carcinoma; OV, ovarian serous cystadenocarcinoma; PAAD, pancreatic adenocarcinoma; STAD, stomach adenocarcinoma; THYM, thymoma; UCEC, uterine corpus endometrial carcinoma.

**Table 1 cancers-15-02801-t001:** Correlation between MAL-family expression and overall survival from reports relying on data distinct from large-scale datasets.

Gene	Cancer	Prognosis ^1^	*p* Value	Technique	References
MAL	COAD-READ	Favorable	<0.05	DNA methyl	[124]
	cHL	Unfavorable	0.002	IHC	[157]
	OV	Unfavorable	0.0004	IHC	[156]
	STAD	Favorable	0.03	DNA methyl	[75]
	STAD	Favorable	<0.05	RT-qPCR	[76]
	UCEC	Unfavorable	<0.05	IHC	[152]
MAL2	COAD-READ	Unfavorable	<0.001	IHC	[102]
	PAAD	Unfavorable	0.03	IHC	[99]
MALL	COAD	Favorable	0.008	IHC	[113]
CMTM8	STAD	Favorable	<0.05	IHC	[117]
MYADML2	LIHC	Unfavorable	0.013	IHC	[150]

^1^ Referred to the high-expressing or the hypermethylated group.

## Data Availability

The results shown here are based upon data generated by the TCGA Research Network (https://www.cancer.gov/tcga) and the GTEx Project (gtexportal.org). The web servers cBioPortal (cbioportal.org), GEPIA2 (gepia2.cancer-pku.cn), Kaplan-Meier plotter (kmplot.com), Human Protein Atlas (HPA, proteinatlas.org), and ROC plotter (rocplot.org) were last accessed on 24 February 2023.

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
