# Peer review of "The MAL Family of Proteins: Normal Function, Expression in Cancer, and Potential Use as Cancer Biomarkers"

_cancers, 2023, doi:10.3390/cancers15102801_

Round 1
Reviewer 1 Report
The review entitled “The MAL Family of Proteins: Normal Function, Expression in Cancer, and Potential Use as Cancer Biomarkers” by L. Labat-de-Hoz et al. describes in detail the MAL protein family, in terms of biochemical features, functions and expression in normal and cancer tissues, by considering and discussing also the genetic/epigenetic lesions, the functions as well as the possible use as therapeutic targets and/or prognostic/diagnostic biomarkers in cancer.
The review is very well organized, and the contents are clearly described. I suggest the following minor modifications to improve the quality of the manuscript:
A recent reference regarding MYADML2 in HCC (Zhang B et al, iScience. 2023 Feb 2;26(3):106099) is available and should be cited in the manuscript.
Figure 4: in addition, maybe it would be useful to provide also a comprehensive table describing the genes’ expression levels in tumors compared to normal tissues.
Paragraph 5: are the MAL family genes regulated by miRNAs? Please, provide a description.
Conclusions and future work: the Authors conclude highlighting the role of MAL family as cancer biomarkers or target antigens for CAR-T therapy. I think that these statements should be softer (e.g. row 779 “as target antigens” should be changed to “as putative target antigens”, row 782 “the routine use” to “the possible routine use”, etc.).
Reviewer 2 Report
Labat de Hoz et al give an overview of the MAL family with a focus on expression of its members in cancers and their potential as biomarkers. Overall the review is clearly written and well-structured and the display items are helpful. The authors have cited over 230 papers which indicators that the review strives to be comprehensive although based on the breadth of the field this is difficult to judge. A relevant paper on MAL (https://doi.org/10.1002/eji.202048603) is not mentioned potentially because it reports data that partially contradict a study of the authors. Specifically the authors of the study did not find evidence for a role of MAL in Lck transport and function which is in line with a lack of MAL expression in murine T cells and also that Lck but not MAL was a highly ranked hit in CRISPR screens for molecules that have an essential role in T cell activation. This results should be mentioned an discussed. Furthermore, in this study it was also shown that in T cells MAL is associated with a non-differentiated non activated phenotype. In their review Labat de Hoz et al report several lines of evidence that MAL expression is associated with better survival in several cancers and that it can act as a tumor suppressor. Is there evidence that MAL downregulation is associated with differentiation and or proliferation also in cancer? Finally the study by Leitner et al also describes a MAL antibody and this antibody should be added to Table 2.
The role of MAL2 as biomarker in breast cancer is interesting – is there a potential for therapeutic antibodies targeting this antigen in breast cancer?
The authors should consider the add a list of abbreviations.
In figure 5 the authors should consider to move the legend (amplification, deletion and mutation) out of the first panel to the free space at the bottom right of the figure.
Figure 7: Maybe I have overlocked something it but is not clear to me what the numbers in the blue and red boxes indicate.
Reviewer 3 Report
The authors provide a thorough review of MAL family proteins and how their expression is altered in cancer. Abbarent Mal family protein expression can affect phenotypes such as ameboid movement of cells or endocytosis of tumor antigens, which in turn can profoundly affect cancer progression and survival. It is a good review and should be published.
Round 2
Reviewer 2 Report
The authors have addressed most of the points raised by this reviewer. However the sentence “…results of a recent study using an antibody to MAL that was not biochemically validated have cast doubt on this role [26].” is misleading since it implies that the antibody used in this study was not properly validated. This is not true since this antibody was identified by expression cloning and it was shown that expression of MAL-cDNA in a mouse cell line confers reactivity of the antibody to this cell line. In addition it was shown that knock out of MAL in Jurkat cells abrogates reactivity of this antibody. Thus the antibody is properly validated and the author should acknowledge that.
In table 2 “KO” and ”expression cloning” should be added to the column “validation” of this antibody.
Round 3
Reviewer 2 Report
The authors have grossly distorted results by a group that are not in line with their work in an earlier review about MAL. Now they try to do it again by insinuating the MAL antibody used by Leitner et al was not properly validated which is not true. This is not acceptable.